# Neuralized Markov Random Field for Interaction-Aware Stochastic Human Trajectory Prediction

**Zilin Fang**[1], **David Hsu**[1,2], **Gim Hee Lee**[1]
[1]School of Computing, National University of Singapore (NUS) [2]Smart Systems Institute, NUS

## Abstract

Interactive human motions and the continuously changing nature of intentions pose significant challenges for human trajectory prediction. In this paper, we present a neuralized Markov random field (MRF)-based motion evolution method for probabilistic interaction-aware human trajectory prediction. We use MRF to model each agent's motion and the resulting crowd interactions over time, hence is robust against noisy observations and enables group reasoning. We approximate the modeled distribution using two conditional variational autoencoders (CVAEs) for efficient learning and inference. Our proposed method achieves state-of-the-art performance on ADE/FDE metrics across two dataset categories: overhead datasets ETH/UCY, SDD, and NBA, and ego-centric JRDB. Furthermore, our approach allows for real-time stochastic inference in bustling environments, making it well-suited for a 30FPS video setting. We open-source our codes at: https://github.com/AdaCompNUS/NMRF_TrajectoryPrediction.git.

## 1 Introduction

Trajectory prediction aims to estimate the most possible future states of one or multiple interacting agents conditioned on past observations. It also serves as the underlying foundation for various intelligent systems, including autonomous driving (Jiang et al., 2023; Girase et al., 2021), human-robot interaction systems such as robot guidance (Aghili, 2012; Rudenko et al., 2020), service robots (Lee et al., 2018; Sun et al., 2018), *etc*. One of the key challenges in this task is to capture the inherent multimodal behaviors of agents, which are influenced by environmental obstacles, social rules, self-intention changes, interactive dynamics, *etc*. Compared to vehicle trajectories, human motions exhibit more short-term uncertainties due to fewer lane constraints in open spaces, and smaller momentum makes the movements more agile and flexible, rendering them difficult to predict.

Current prediction works mainly address such multi-agents entangling problems from two aspects: i) stochastic algorithms that generate multiple future trajectories simultaneously (Gupta et al., 2018; Salzmann et al., 2020; Gu et al., 2022) to cover possible intention changes; ii) interaction modeling through graphs at various scales (Li et al., 2020; Xu et al., 2022a), socially-aware operations (Alahi et al., 2016; Xu et al., 2018), attention mechanism (Yu et al., 2020; Girgis et al., 2021), *et al*. Furthermore, these two directions are usually combined for further performance improvements (Mangalam et al., 2020; Sadeghian et al., 2019; Yuan et al., 2021). Most of these methods assume that the interaction pattern can be captured and applied to future time, and thus extract human interaction features only from history sequences. Some other works such as latent self-motion (Choi & Hebert, 2006), intended actions with heuristics (Han et al., 2023), crowds as dissipative systems modeling (Bhaskara et al., 2024), *etc.* apply the Markov property in their framework designs to construct temporal and spatial dependencies. Our method differs with an explicit structured modeling of each agent's motion and crowd interactions throughout the entire sequences, essentially state dynamics, employing MRF to iteratively infer the stochastic distribution of future joint configurations.

In this work, we introduce an MRF realized by neural networks for interaction-aware human trajectory prediction, as depicted in Figure 1. We first derive the crowd motion evolution as a probabilistic distribution, which consists of: i) a **Bayesian update term** to predict the next states based on the given observations; and ii) a **transition term** that is further factorized into a **self-evolution term** to

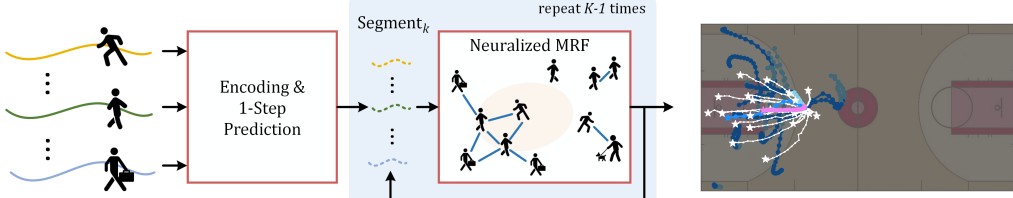

Figure 1: Neuralized MRF for motion and interaction aware trajectory prediction. Red boxes represent computation modules. Our proposed method leverages Markovian motion assumption and explicitly models future transitions and interactions. Through iterative prediction, it produces a concentrated distribution around the ground truth with more accurate best-of-20 predictions.

predict the next states based on their current states, and an **interaction term** to model the spatial relations between agents. Figure 2 sketches the proposed MRF-based evolution. It assumes that short-term human motions are Markovian up to certain frequencies. By dividing the future joint space configuration (*i.e.*, the collection of individual trajectories) sequences into multiple periods, the resulting trajectory segments form a Markov chain conditioned on past observations. Within each segment, trajectory clips are treated as nodes in a dynamically constructed graph, with edges linked based on a distance threshold to model interactions between nearby agents. Consequently, a joint configuration state can be represented by either one undirected graph or a set of multiple undirected sub-graphs. The internal motion dynamics of the state are explicitly modeled by the MRF, which inherently captures the evolving pair-wise and group-wise relationships for the future.

We approximate the distribution using a neural network framework with two CVAEs, employing variational inference as exact inference is intractable. Our proposed framework is composed of three sub-networks: i) a CVAE for the Bayesian update term; ii) a recurrent CVAE for the transition term comprising the self-evolution and interaction terms; and iii) a neural sampler to generate multimodal samples from the learned latent distributions. We conduct experiments across four datasets named ETH/UCY, SDD, NBA SportVU, and JRDB to validate the effectiveness of our approach. Through quantitative and qualitative results, we show that our proposed neural interaction-aware MRF can capture dynamic and hierarchical interactions among agents via neuralized potentials for future behaviors. Our approach achieves state-of-the-art performance under ADE/FDE metrics, while yielding real-time inference speed across various environments at the same time. Furthermore, our method is robust across the simulated observation noise, and the learned potentials in MRF can be utilized for additional human-centered scene understanding tasks such as group inference.

Our contributions can be summarized as follows: i) A new MRF-based framework for the human trajectory prediction task. Our MRF explicitly models the agent's motion dynamics and the resulting crowd interactions. ii) Tractable learning and inference of the MRF by introducing a novel and lightweight neural network. iii) The state-of-the-art and time-efficient prediction performance of our approach, along with its robustness under noise disturbance, is demonstrated through evaluations on interaction-rich datasets.

## 2    RELATED WORKS

**Interaction Modeling.**    Previous works have shown various approaches to model the interactions within crowds, utilizing RNNs (Vemula et al., 2018; Zhang et al., 2019), graph networks (Mohamed et al., 2020; Salzmann et al., 2020; Xu et al., 2022a), attention mechanisms (Kosaraju et al., 2019; Mangalam et al., 2020; Kamra et al., 2020), and transformers to capture spatial and temporal dependencies together (Yu et al., 2020; Giuliari et al., 2021; Yuan et al., 2021; Saadatnejad et al., 2023; Shi et al., 2023). Specifically, graph-based networks usually aggregate edge information from neighboring agents and conduct message passing to obtain the final ego-node representations (Salzmann et al., 2020; Hu et al., 2020); while attention and transformer structures attribute varying importance levels for spatially proximal agents and focus on the most relevant ones to assist social-influence counting. Our approach explicitly models the full dynamics of joint space state transitions and interactions with an MRF, and attentions are employed to extract features from the past.

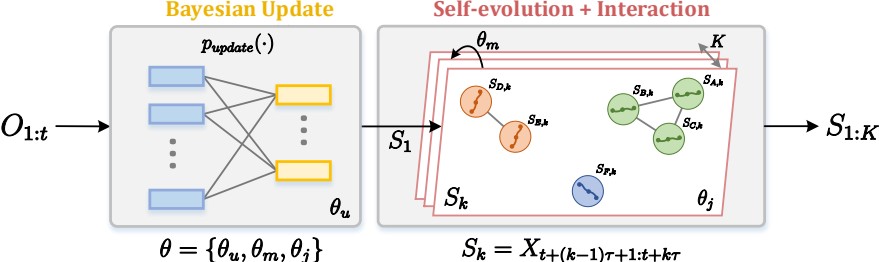

Figure 2: Overview of our MRF-based human motion evolution. Given the history sequence, $O_{1:t}$, the first future segment $S_1$ is estimated via the Bayesian update module, and future segments $S_{2:K}$ are generated from the self-evolution and interaction module. The Bayesian update, self-evolution and interaction modules are parameterized by $\theta = \{\theta_u, \theta_m, \theta_j\}$.

**Stochastic Trajectory Prediction.** Aside from deterministic methods (Alahi et al., 2016; Vemula et al., 2018; Zhang et al., 2019), more stochastic trajectory prediction works with deep generative models have been proposed to manage the multi-modalities of human motions. Generative Adversarial Network (GAN) (Gupta et al., 2018; Sadeghian et al., 2019; Dendorfer et al., 2021), Variational Auto-Encoder (VAE) (Mangalam et al., 2020; Lee et al., 2022; Xu et al., 2022c), and Conditional Variational Autoencoder (CVAE) (Lee et al., 2017; Salzmann et al., 2020; Cheng et al., 2021; Xu et al., 2022a; Yue et al., 2022; Mohamed et al., 2020) are several lines, with some variants integrating normalizing flow to implement non-Gaussian priors (Bhattacharyya et al., 2019) or heatmap to utilize rasterized images (Mangalam et al., 2021). Recently, denoising diffusion probabilistic models have also been applied into time series forecasting (Rasul et al., 2021), thereby expanding to trajectory prediction. MID (Gu et al., 2022) formulates the trajectories generation process as a reverse diffusion process by Markov chain with Gaussian transitions. To tackle the sample generation efficiency, LED (Mao et al., 2023) utilizes a learned initializer to produce correlated samples in the reverse process, replacing the Gaussian prior and skipping numerous denoising steps. SingularTrajectory (Bae et al., 2024b) embeds all types of motion patterns into a Singular space for the generality across benchmarks and uses cascaded denoising in its diffusion-based predictor, significantly surpassing MID.

**Markov Random Field Applications.** As a popular tool for modeling dependent distributions with an undirected graph, the Markov random field (MRF) has been widely applied in tasks such as multi-target tracking (Khan et al., 2004) and location inferring (Mattyus et al., 2015), among others. Recently, NMRF (Guan et al., 2024) has also utilized MRFs to capture complicated pixel relationships for stereo matching. In trajectory prediction tasks, the Markov property-*i.e.*, a pedestrian's future state primarily depends only on its current state-has been implemented for agent-wise self-motion modeling, from early piecewise segment approach (Choi & Hebert, 2006) to more recent FlowMNO (Bhaskara et al., 2024). By using a heuristic that relates coordinates and actions, such as turning right or left, S-T CRF (Han et al., 2023) employs a CRF to model intended actions for future timesteps. Rather than applying the Markov property only to sub-components, we use the MRF to model the full dynamics of state transitions and crowd interactions, providing a neural network-based method for learning and inference.

## 3 CROWDS MOTION EVOLUTION WITH MARKOV RANDOM FIELD

### 3.1 PROBLEM FORMULATION

Given $t$ frames of observations $O_{1:t} \triangleq \{O_1, O_2, ..., O_t\}$ up to the current time $t$, the goal of human trajectory prediction is to determine the future joint configuration space of the humans $X_{t+1:t+T} \triangleq \{X_{t+1}, X_{t+2}, ..., X_{t+T}\}$ in the next $T$ time steps. We further define the joint space configuration $X_t \triangleq \{X_{i,t}\}_{i=1}^M$ to consist of $M$ individual states.

Although it is possible to solve the human trajectory prediction problem with a deterministic function map $f : O_{1:t} \mapsto X_{t+1:t+T}$, we model a probabilistic distribution $p(X_{t+1:t+T} \mid O_{1:t})$ to en-

hance the learning of human interactions under noisy observations for future trajectory prediction. Formally, we define the evolution of joint configuration segments as $S = \{S_1, \cdots, S_K \mid S_k \triangleq X_{t+(k-1)\tau+1:t+k\tau}\}$. $S$ is a Markov chain parameterized by $\theta$ to indicate that the future prediction segments with an assumed period $\tau$ are conditionally independent of the history segments given the current state and transition dynamics hyper-parameter. As a result, we get:

$$
\begin{aligned}
p(X_{t+1:t+T} \mid O_{1:t}, \theta) &= p(X_{t+1}, X_{t+2}, ..., X_{t+T} \mid O_{1:t}, \theta) \\
&= p(S_1, S_2, ..., S_K \mid O_{1:t}, \theta) \\
&= p(S_K \mid S_{1:K-1}, O_{1:t}, \theta)p(S_{1:K-1} \mid O_{1:t}, \theta) \\
&= p(S_K \mid S_{1:K-1}, O_{1:t}, \theta) \ldots p(S_2 \mid S_1, O_{1:t}, \theta)p(S_1 \mid O_{1:t}, \theta) \\
&= p(S_K \mid S_{K-1}, O_{1:t}, \theta) \ldots p(S_2 \mid S_1, O_{1:t}, \theta)p(S_1 \mid O_{1:t}, \theta) \\
&= \underbrace{p(S_1 \mid O_{1:t}, \theta)}_{\text{Bayesian update}} \underbrace{\prod_{k=1}^{K-1} p(S_{k+1} \mid S_k, \theta)}_{\text{self-evolution + interaction}},
\end{aligned}
\tag{1}
$$

where $K = |S| = T//\tau$ is the set size of configuration segments determined by period $\tau$ (also referred to as stride in network implementation). $\theta$ is typically estimated from observations which encodes Bayesian update $\theta_u$ based on past measurements, and system dynamics that include individual predictive motion models $\theta_m$ and pairwise interactions $\theta_j$, with visualizations in Figure 2.

### 3.2 REALIZATION WITH CONDITIONAL VARIATIONAL AUTOENCODER

We propose to realize the posterior distribution $p(X_{t+1:t+T} \mid O_{1:t}, \theta)$ with conditional variational autoencoders (CVAEs) since exact inference is computationally intractable. Let $Z$ represent the latent embedding of the CVAE. We define the log posterior as: $\log p(S_1, \cdots, S_K \mid O_{1:t}, \theta) \triangleq \log p_\theta(S \mid O)$ and maximize the log-posterior with the evidential lower bound:

$$
\begin{aligned}
\log p_\theta(S \mid O) &\geq \mathcal{L}_{ELBO} \\
&= \mathbb{E}_{q_\psi(Z|O,S)}\big[\log p_\phi(S \mid Z, O)\big] - D_{\mathrm{KL}}(q_\psi(Z \mid O, S) \parallel p(Z)),
\end{aligned}
\tag{2}
$$

where $p_\phi(\cdot)$ is the decoder and $q_\psi(\cdot)$ is the encoder parameterized by $\phi$ and $\psi$, respectively. The first term is the reconstruction loss and the second term enforces the latent space to be close to the normal distribution. $S$ is the reconstructed output of the decoder and $O$ is the condition of the CVAE. We further assume that the latent embedding $Z$ is conditionally independent of $O$ given $S$, i.e., $q_\psi(Z \mid O, S) = q_\psi(Z \mid S)$. The CVAE can be trained by minimizing the evidential lower bound on an $n$-item dataset $\{\mathbb{O}, \mathbb{X}\}$, where $|\mathbb{O}| = |\mathbb{X}| = n$, $\mathbb{O} = \{O_{1:t}\}$ is the set of observations and $\mathbb{X} = \{X_{t+1:t+T}\}$ is the set of ground truth predictions (*cf.* Section 3.4 more the training details).

### 3.3 NETWORK ARCHITECTURE

Figure 3 shows an illustration of our network architecture that consists of two CVAEs: 1) *Bayesian Update* and 2) *MRF-based Evolution* to approximate the two factors of the posterior distribution derived in Equation 1.

**Bayesian Update CVAE.** We design the Bayesian Update CVAE to approximate the Bayesian update factor $p(S_1 \mid O_{1:t}, \theta) = p(S_1 \mid O_{1:t}, \theta_u)$ in the posterior distribution $p_\theta(S \mid O)$. The given observations $O_{1:t}$ are first encoded into time-dependent features using the History Encoder constructed from convolutional layers and the gated recurrent units (GRU), and then fed into a single self-attention layer to output socially-dependent features. We use the socially-dependent features as the condition for our Bayesian Update CVAE. We introduce a Future Encoder which acts as the CVAE encoder to encode the given predicted trajectory $X_{t+1:t+T}$ into the latent embedding $Z \sim \mathcal{N}(\mu, \sigma^2)$ during CVAE training. Finally, we concatenate the socially-dependent features with the latent embedding as the input to a multilayer perceptron-based Update Decoder that reconstructs the first segment $\hat{S}_1$. Note that the Future Encoder and the Update Decoder play the role of the CVAE encoder $q_\psi(Z \mid S)$ and decoder $p_\phi(S \mid Z, O)$ in Equation 2, respectively.

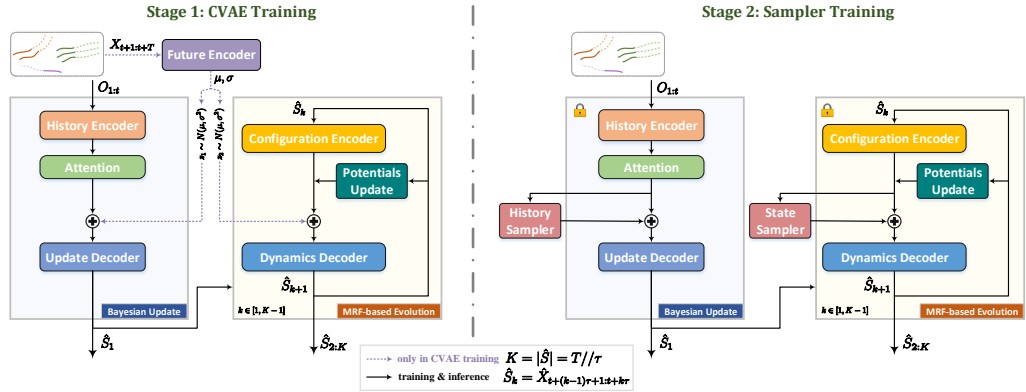

Figure 3: Our network architecture. The Bayesian Update module generates predictions $\hat{S}_1$ for the first defined sequence segment, which then functions as the input into the MRF-based Evolution module to produce succeeding samples $\hat{S}_2 : \hat{S}_K$ iteratively.

**MRF-based Evolution CVAE.** We propose the MRF-based Evolution CVAE to approximate the self-evolution and interaction factor $\prod_{k=1}^{K-1} p(S_{k+1} \mid S_k, \theta)$ in the posterior distribution $p_\theta(S \mid O)$. We further factorize $p(S_{k+1} \mid S_k, \theta)$ into two parts:

i) The ***space-level distribution*** $p(S_k \mid \theta_j)$ which models the $k$-th joint space configuration. We are inspired by (Khan et al., 2004; 2005) to construct a Markov random field (MRF) with an undirected graph $G = (V, E)$, where the segments $S$ are the nodes $V$ and the edges $E$ represent the interaction between the segments, *i.e.*:

$$p(S_k \mid \theta) = p(S_k \mid \theta_j) \propto \prod_{ij \in \mathrm{E}} \gamma(S_{i,k}, S_{j,k} \mid \theta_j). \tag{3}$$

$\gamma(\cdot)$ is the pairwise potential of the MRF. We link the edges $E$ dynamically according to pairwise distance in each computation step.

ii) The ***agent-level distribution*** $p(S_{i,k+1} \mid S_{i,k}, \theta_m)$ which models the agent-wise self-transition. Putting the space-level and agent-level distributions together, the self-evolution and interaction factor in Equation 1 becomes:

$$p(S_{k+1} \mid S_k, \theta) \propto \prod_i p(S_{i,k+1} \mid S_{i,k}, \theta_m) \prod_{ij \in \mathrm{E}} \gamma(S_{i,k}, S_{j,k} \mid \theta_j), \tag{4}$$

which we approximate with the MRF-based Evolution CVAE. To this end, we propose a Configuration Encoder which predicts $S_{i,k+1}$ from $S_{i,k}$ to approximate $p(S_{i,k+1} \mid S_{i,k}, \theta_m)$. We further introduce a Potential Update module which calculates pairwise spatial distances by using the last frame of the configuration segment, links edges according to a defined distance threshold $d$, and accumulates edge features into the connected nodes to approximate the MRF $\prod_{ij \in \mathrm{E}} \gamma(S_{i,k}, S_{j,k} \mid \theta_j)$.

The two features from the Configuration Encoder and Potential Update module are the condition of the CVAE, which are then concatenated with the latent embedding $Z$ and fed into the Dynamics Decoder to output the joint space configuration $\hat{S}_{k+1}$ for the next timestep. These procedures are looped until the final segment prediction is obtained. All sub-modules are implemented using multilayer perceptions. Note that the Future Encoder and the Dynamics Decoder play the role of the CVAE encoder $q_\psi(Z \mid S)$ and decoder $p_\phi(S \mid Z, O)$ in Equation 2, respectively.

### 3.4 NETWORK TRAINING

We train our network in two stages as illustrated in Figure 3. In Stage 1, we train the CVAEs using the evidential lower bound loss derived in Equation 2. The reconstruction loss is defined as the

minimum error across $N$ samples and the KL-divergence is summed over all segments as follows:

$$\mathcal{L}_{stage1} = \alpha \cdot \min_N ||\hat{X}_{t+1:t+T} - X_{t+1:t+T}||_2 + \beta \cdot \sum_\tau D_{\text{KL}}(q_\phi \| \mathcal{N}(0, I)). \quad (5)$$

Instead of directly sampling from the standard normal distribution $\mathcal{N}(0, I)$, we further train two samplers: a History Sampler and a State Sampler with other modules frozen in Stage 2 for purposive sampling as illustrated in the right-half of Figure 3, where we replace the KL-divergence with discrepancy loss (Bae et al., 2022b) of samples over the encoded joint configuration state:

$$\mathcal{L}_{stage2} = \alpha \cdot \min_N ||\hat{X}_{t+1:t+T} - X_{t+1:t+T}||_2 + \lambda \cdot \frac{1}{N} \sum_i - \log \min_{\substack{j \in [1,\dots,N] \\ j \neq i}} ||\hat{Y}_i - \hat{Y}_j||, \quad (6)$$

where $\hat{Y}_i$ is the $i$-th sampler output among $N$ samples before feature fusion in Stage 2. $\alpha, \beta$ and $\lambda$ are hyperparameters to balance the loss terms during training. After training, we only apply the components in Stage 2 for inference.

## 4 EXPERIMENTS

**Datasets.** We evaluate our methods on four trajectory prediction datasets: ETH/UCY (Pellegrini et al., 2009; Lerner et al., 2007), Stanford Drone Dataset (SDD) (Robicquet et al., 2016), NBA SportVU Dataset (NBA), and the JackRabbot Dataset and Benchmark (JRDB) (Martin-Martin et al., 2021), covering interaction-rich indoor and outdoor scenarios. **ETH/UCY** dataset contains five subsets: ETH, HOTEL, UNIV, ZARA1, and ZARA2. We follow the leave-one-out approach from (Gupta et al., 2018) with four subsets for training-validation and the remaining subset for testing, predicting the future 12 frames (4.8s) using 8 frames observations (3.2s). **SDD** is a bird's eye view pedestrian dataset collected on a university campus. It is originally annotated in pixels without providing precise projection matrices. We predict the future 12 frames (4.8s) on 8 frames (3.2s) history and report the metrics on both pixel and meter units. **NBA** contains the trajectories of 10 players and the ball in real basketball games collected by the SportVU tracking system. We predict the future 20 frames (4.0s) based on 10 frames (2.0s) history, similar to most other works. The agent's intention changes suddenly and frequently in this dataset, resulting in zigzags, sharp turnarounds, *etc* which increases difficulties compared to the pedestrian datasets. **JRDB** is a large-scale egocentric dataset recorded by a social robot in indoor and outdoor environments with stationary and moving behaviors. We follow the train-validation-test split applied in Social-Transmotion (Saadatnejad et al., 2023) for deterministic prediction. For stochastic situations, splits are in accordance with the official JRDB detection and tracking challenge. Since all labels contained are in camera view and the robot is moving for several scenarios, we utilize the provided odometry information in rosbags to transform all trajectories into global world coordinates. As officially suggested, the trajectory of future 12 frames is predicted over the past 9 frames with 2.5 frames per second frequency.

**Implementation details.** We calculate displacement for each configuration segment instead of absolute quantities for robustness: $\hat{S}_{k+1} = \hat{S}_k + \Delta\hat{S}_k$, where $\Delta\hat{S}_k$ is network output. As mentioned in Section 3.3, the History Encoder comprises a 1D convolution layer with a kernel size of three and an output channel of 32, alongside a GRU layer with a hidden embedding dimension of 256, similar to LED (Mao et al., 2023). All the other modules are built using multi-layer perceptions, and the distance threshold $d$ for the potential update module is set to 3m. We set sample number $N$ to 20 in both training and testing stages for stochastic prediction, and train the entire model with AdamW optimizer and StepLR scheduler on one Quadro RTX 8000 GPU. The latent vectors $Z_1$ and $Z_2$ have 16 dimensions and the embedding dimension of the configuration encoder is 32. Other network details and dataset-specific hyperparameters are attached in the appendix.

**Evaluation metrics.** We evaluate our proposed method with ADE and FDE for a single sample in the deterministic case, and with $\text{minADE}_{20}$ and $\text{minFDE}_{20}$ in the stochastic cases following the previous works (Mangalam et al., 2020; Mao et al., 2023). ADE quantifies the average displacement error between predictions and ground truth across all time steps, while FDE measures the displacement error from the final timestep to the current time. The $\text{minADE}_{20}$ and $\text{minFDE}_{20}$ metrics calculate the minimum error among 20 predictions, and it is commonly used in evaluating generative models' ability to generate high-quality samples.

Table 1: $minADE_{20}/minFDE_{20}$ in **meters** on ETH/UCY datasets. **Bold**/underline indicate the best/second-best results. * denotes results reproduced from official codes. † marks results where the best model was chosen directly based on the test set. We address this issue by following the standard train-val-test split in Social-GAN (Gupta et al., 2018), selecting the best model via the validation set. ‡ is for the future information leakage issue fixed. Inference speed is measured for generating 20 samples under a scene with 57 pedestrians.

| Subset | Social-GAN* (Gupta et al., 2018) | Social-STGCNN (Mohamed et al., 2020) | PECNet*† (Mangalam et al., 2020) | STAR (Yu et al., 2020) | Trajectron++*‡ (Salzmann et al., 2020) | SGCN (Shi et al., 2021) | AgentFormer* (Yuan et al., 2021) | NPSN*(best) (Bae et al., 2022b) | GP-Graph (Bae et al., 2022a) | MID*†(DDIM) (Gu et al., 2022) | TUTR*† (Shi et al., 2023) | SingularTrajectory* (Bae et al., 2024b) | SocialCircle*† (Wong et al., 2024) | Ours |
|---|---|---|---|---|---|---|---|---|---|---|---|---|---|---|
| ETH | 0.77/1.40 | 0.65/1.10 | 0.64/1.13 | 0.36/0.65 | 0.61/1.03 | 0.63/1.03 | 0.46/0.73 | 0.37/0.60 | 0.43/0.63 | 0.46/0.74 | 0.45/0.67 | 0.35/0.46 | 0.27/0.42 | **0.26/0.37** |
| HOTEL | 0.43/0.88 | 0.50/0.86 | 0.22/0.38 | 0.17/0.36 | 0.20/0.28 | 0.32/0.55 | 0.14/0.23 | 0.16/0.25 | 0.18/0.30 | 0.18/0.30 | 0.14/0.20 | 0.13/0.20 | 0.13/0.16 | **0.11/0.17** |
| UNIV | 0.74/1.50 | 0.44/0.80 | 0.35/0.57 | 0.31/0.62 | 0.30/0.54 | 0.37/0.70 | 0.25/0.44 | **0.23/0.39** | 0.24/0.42 | 0.25/0.48 | 0.24/0.44 | 0.27/0.47 | 0.29/0.51 | 0.28/0.49 |
| ZARA1 | 0.35/0.70 | 0.34/0.53 | 0.25/0.45 | 0.26/0.55 | 0.24/0.41 | 0.29/0.53 | 0.18/0.30 | 0.18/0.32 | 0.17/0.31 | 0.23/0.45 | 0.19/0.36 | 0.19/0.33 | 0.19/0.33 | **0.17/0.30** |
| ZARA2 | 0.36/0.72 | 0.31/0.48 | 0.18/0.31 | 0.22/0.46 | 0.17/0.32 | 0.25/0.45 | **0.14/0.24** | 0.14/0.25 | 0.15/0.29 | 0.18/0.35 | 0.15/0.28 | 0.15/0.27 | 0.14/0.25 | 0.14/0.24 |
| AVG (meters) | 0.53/1.04 | 0.45/0.75 | 0.33/0.60 | 0.26/0.53 | 0.30/0.52 | 0.37/0.65 | 0.23/0.39 | 0.22/0.36 | 0.23/0.39 | 0.26/0.46 | 0.23/0.39 | 0.22/0.34 | 0.20/0.33 | **0.19/0.32** |
| Speed (57 per.) | ∼ 63.2ms | ∼ 1.8ms | ∼ 47.6ms | ∼ 2376ms | ∼ 254ms | ∼ 6.5ms | ∼ 108ms | ∼ 8.6ms | ∼ 30ms | ∼ 400ms | ∼ 591ms | ∼ 15.3ms | ∼ 17.4ms | ∼ 9.8ms |

Table 2: $minADE_{20}/minFDE_{20}$ in **pixels** and **meters** (if available) on SDD dataset. **Bold**/underline indicate the best/second-best results. * denotes results reproduced from official codes. Train-test splits are the same as the baseline Social-VAE (Xu et al., 2022c).

| | Social-GAN (Gupta et al., 2018) | Social-STGCNN (Mohamed et al., 2020) | PECNet (Mangalam et al., 2020) | Trajectron++ (Salzmann et al., 2020) | BiTraP (Yao et al., 2021) | MID* (Gu et al., 2022) | MemoNet (Xu et al., 2022b) | SGNet-ED (Wang et al., 2022) | GP-Graph (Bae et al., 2022a) | Social-VAE (Xu et al., 2022c) | Social-VAE+FPC (Xu et al., 2022c) | TUTR* (Shi et al., 2023) | SocialCircle* (Wong et al., 2024) | Ours |
|---|---|---|---|---|---|---|---|---|---|---|---|---|---|---|
| pixels (meters if avl.) | 27.23/41.44 | 20.8/33.2 | 9.29/15.93 | 10.00/17.15 (0.34/0.58) | 9.09/16.31 (0.32/0.57) | 9.08/17.04 | 8.56/12.66 | 9.69/17.01 | 9.1/13.8 (0.33/0.58) | 8.88/14.81 (0.30/0.50) | 8.10/11.72 (0.27/0.39) | 7.90/12.96 | 10.11/16.53 | **7.20/11.29** (0.25/0.39) |

## 4.1 TRAJECTORY PREDICTION RESULTS

**ETH/UCY.** Table 1 shows the comparison between our model and recent state-of-the-art benchmarks on the ETH/UCY dataset. Since several baselines report their models' test performances by confounding the validation set with the test split, we correct these numbers by re-training with official codes and selecting the best model based on the validation set to ensure fairness. Additionally, recent work such as LMTraj (Bae et al., 2024a) utilizes large language models (LLMs) to capture the multi-modality of human trajectories. However, we omit LMTraj from the baseline since its inference speed is heavily constrained by the response times of models such as GPT-3.5 and GPT-4, and its prediction performance does not surpass SingularTrajectory (Bae et al., 2024b). As shown in the table, our method balances the prediction accuracy and computational efficiency with the best average $minADE_{20}/minFDE_{20}$ and an inference speed faster than 100Hz under a 57-pedestrian scenario in the ETH/UCY dataset. It outperforms diffusion-based methods like MID (Gu et al., 2022) and SingularTrajectory (Bae et al., 2024b), and the previous SOTA SocialCircle (Wong et al., 2024).

**SDD.** Table 2 shows the comparisons on the SDD dataset with our proposed model maintaining the best performance. We transform the pixel-unit annotations into meter-unit following the baseline SocialVAE (Xu et al., 2022c) since our method uses a meter-based distance threshold $d$. It is important to note that due to the lack of precise projection matrices in the SDD dataset, the pixel-to-meter scale is estimated with large variation and unreliability with some values derived from Google Maps and others based on rough guesses (Amirian et al., 2020). We include comparisons on this dataset because it is widely used in the community. However, we must emphasize that pixel errors would only be meaningful if a pixel corresponds to a fixed physical distance across all scenes, which unfortunately is not guaranteed by SDD.

**NBA.** As presented in Table 3, the proposed method is then evaluated against 10 baselines on the NBA dataset. Our method maintains state-of-the-art accuracy to the best-of-20 samples. We compare the inference time with several baselines where official implementations for this dataset are provided, showing that our model infers trajectories less than one-third of the time while preserving the best prediction performance. Figure 4 visualizes the estimation results from the current best

Table 3: $minADE_{20}/minFDE_{20}$ in **meters** on NBA dataset. **Bold**/underline indicate the best/second-best results. * denotes results reproduced from official codes.

| Time | Social-GAN (Gupta et al., 2018) | Social-STGCNN (Mohamed et al., 2020) | PECNet (Mangalam et al., 2020) | STAR (Yu et al., 2020) | Trajectron++ (Salzmann et al., 2020) | MemoNet (Xu et al., 2022b) | NPSN (Bae et al., 2022b) | GroupNet (Xu et al., 2022a) | MID (Gu et al., 2022) | LED* (Mao et al., 2023) | Ours |
|---|---|---|---|---|---|---|---|---|---|---|---|
| 1.0s | 0.41/0.62 | 0.34/0.48 | 0.40/0.71 | 0.43/0.66 | 0.30/0.38 | 0.38/0.56 | 0.35/0.58 | 0.26/0.34 | 0.28/0.37 | 0.18/0.27 | **0.16/0.24** |
| 2.0s | 0.81/1.32 | 0.71/0.94 | 0.83/1.61 | 0.75/1.24 | 0.59/0.82 | 0.71/1.14 | 0.68/1.23 | 0.49/0.70 | 0.51/0.72 | 0.37/0.56 | **0.34/0.50** |
| 3.0s | 1.19/1.94 | 1.09/1.77 | 1.27/2.44 | 1.03/1.51 | 0.85/1.24 | 1.00/1.57 | 1.01/1.76 | 0.73/1.02 | 0.71/0.98 | 0.58/0.85 | **0.53/0.75** |
| Total(4.0s) | 1.59/2.41 | 1.53/2.26 | 1.69/2.95 | 1.13/2.01 | 1.15/1.57 | 1.25/1.47 | 1.31/1.79 | 0.96/1.30 | 0.96/1.27 | 0.82/1.15 | **0.75/0.97** |
| Speed (11 per.) | NA | NA | NA | NA | NA | NA | NA | ∼ 21.4ms | >900ms | ∼ 65.7ms | ∼ 19.3ms |

Samples generated with LED (Mao et al., 2023)

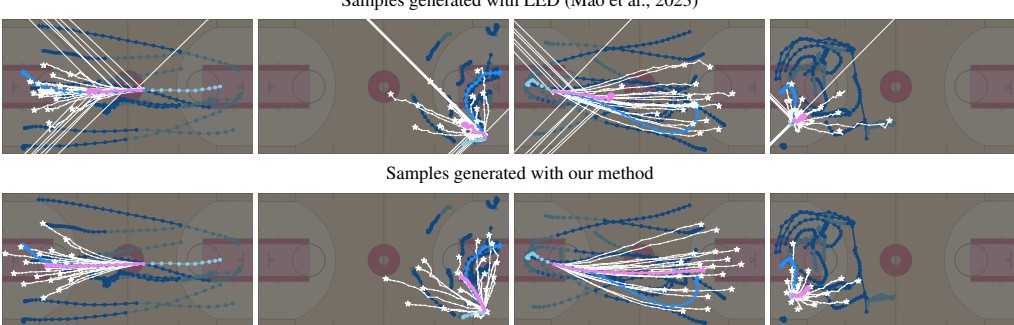

Samples generated with our method

Figure 4: Generated prediction samples with $N = 20$ on NBA dataset. The light blue and dark blue represent the history and future ground truth respectively, and white curves ✦—✦ are samples. Mean estimation is highlighted by violet.

approach and our model. The prediction samples from LED (Mao et al., 2023) are noisy with several sampled positions deviating significantly from the real potential distribution of human motions. This noise can negatively impact downstream tasks requiring estimated distributions with confidence since it is impractical to measure the ground truth future and report the minimum error in such cases.

Table 4: Comparisons for ADE and FDE on JRDB dataset with deterministic trajectory prediction, in **meter** unit. In input modality, 'T' represents trajectory only, and '2d BB' means the bounding box of the person in 2D images. ⋆ highlights previously reported best performance.

| Time | Social-LSTM (Alahi et al., 2016) | Social-GAN (Gupta et al., 2018) | Directional-LSTM (Kothari et al., 2021) | Trajectron++ (Salzmann et al., 2020) | Autobots (Girgis et al., 2021) | EqMotion (Xu et al., 2023) | Social-Transmotion⋆ (Saadatnejad et al., 2023) | Social-Transmotion⋆ (Saadatnejad et al., 2023) | LED ($N$=1) (Mao et al., 2023) | **Ours** ($N$=1) *Stage 1 Only* |
|---|---|---|---|---|---|---|---|---|---|---|
| Input Modality | T | T | T | T | T | T | T | T + 2d BB | T | T |
| ADE/FDE | 0.47/0.95 | 0.50/0.99 | 0.45/0.87 | 0.40/0.78 | 0.39/0.80 | 0.42/0.78 | 0.40/0.77 | 0.37/0.73 | 0.32/0.54 | **0.26/0.48** |

Table 5: Stochastic trajectory prediction comparisons on JRDB dataset. minADE$_{20}$ and minFDE$_{20}$ in **meters** are reported. Numbers are related to a constant reference frame.

| Method | 1.2s | 2.4s | 3.6s | Total(4.8s) | Speed (80 per.) |
|---|---|---|---|---|---|
| LED (Mao et al., 2023) | 0.05/0.07 | 0.09/0.14 | 0.14/0.21 | 0.18/0.28 | $\sim$ 118ms |
| Ours (Stage 1 Only) | 0.05/0.06 | 0.09/0.13 | 0.13/0.20 | 0.17/0.27 | $\sim$ 6.7ms |
| Ours | **0.04/0.05** | **0.08/0.11** | **0.11/0.17** | **0.15/0.23** | $\sim$ 6.8ms |

**JRDB.** We finally explore the performance of our method on JRDB, which contains diverse captured indoor and outdoor environments close to humans by a robot. To ensure a fair comparison with other state-of-the-art methods, many of which provide deterministic results, we degenerate our method to output only one sample at a time. Throughout training, validation, and testing, we set $N$ to 1 for the sampler, and the resulting ADE/FDE metrics (in the world frame) are detailed in Table 4. Leveraging trajectory-only inputs, our approach outperforms the previous best model Social-Transmotion (Saadatnejad et al., 2023) that combines 2D bounding boxes extracted from images by approximately 29% and 34% in terms of ADE and FDE, respectively. Note that trajectory prediction is usually conducted in a consistent reference frame, *i.e.*, the world frame. However, since JRDB is captured by a moving robot, baseline methods may directly use its instantaneous coordinate frame, causing human motion to be coupled with robot movement. Nonetheless, the numbers related to the moving camera frame are attached for reference: ADE/FDE on LED is 0.36/0.69, and ours is 0.33/0.63.

We also conduct experiments under stochastic conditions to further investigate the effectiveness of our proposed approach with results presented in Table 5. Our model significantly outperforms the second-best model LED in deterministic cases in terms of minADE$_{20}$ and minFDE$_{20}$ metrics. Moreover, the inference time of our method is over 17 times faster in a bustling environment involving 80 persons, which indicates it can applied for the standard video setting, and enables real-time capabilities on physical robotic systems or desktop GPUs. Figure 5 illustrates eight examples of our prediction results, demonstrating the ability of our learned distribution to capture human intention changes like sharp turns, sudden stops, *etc*.

Table 6: Robustness test for imperfect past trajectories on JRDB dataset. History trajectory includes 9 frames with a scale equal to one, and the percentage stands for the ratio of test data with noise.

| Noise Type | ADE | FDE |
|---|---|---|
| Gaussian Noise (100%, scale = 0.1) | 0.27 (4 frames) / 0.28 (9 frames) | 0.49 (4 frames) / 0.50 (9 frames) |
| Gaussian Noise (100%, scale = 0.2) | 0.32 (4 frames) / 0.34 (9 frames) | 0.55 (4 frames) / 0.57 (9 frames) |
| Gaussian Noise (100%, scale = 0.5) | 0.71 (4 frames) / 0.88 (9 frames) | 1.18 (4 frames) / 1.44 (9 frames) |
| Dropped History (20%, 5 frames) | 0.26 | 0.48 |
| Dropped History (50%, 5 frames) | 0.26 | 0.49 |
| Dropped History (50%, 7 frames) | 0.29 | 0.52 |
| Dropped History (80%, 7 frames) | 0.30 | 0.54 |
| **Ours (full performance)** | **0.26** | **0.48** |

Table 7: Ablations on sampler types (Left) and different strides (Right). $\text{minADE}_{20}$ and $\text{minFDE}_{20}$ for the entire trajectory are reported. Influences from stride $\tau$ are shown with ZARA1 subset, and they differ across datasets. 'AVG.' is for the average of distance errors on 20 samples and 'SD.' represents the standard deviation. Speed is measured for an 11-person scenario.

| Dataset | Stage 1 Only | Stage 2 Only | Two Stages |
|---|---|---|---|
| ETH | 0.28/0.46 | 0.24/0.40 | 0.26/0.37 |
| HOTEL | 0.13/0.21 | 0.11/0.18 | 0.11/0.17 |
| UNIV | 0.32/0.58 | 0.28/0.48 | 0.28/0.49 |
| ZARA1 | 0.20/0.36 | 0.20/0.33 | 0.17/0.30 |
| ZARA2 | 0.16/0.30 | 0.15/0.26 | 0.14/0.25 |
| SDD (pixels) | 8.16/13.37 | 7.83/12.59 | 7.20/11.29 |
| NBA | 0.83/1.16 | 0.77/0.98 | 0.75/0.97 |

| Stride $\tau$ | $\text{minADE/} \text{FDE}_{20}$ | AVG. | SD. | Speed ($\sim$) |
|---|---|---|---|---|
| 1 | 0.28/0.35 | 0.84/1.66 | 0.44/0.92 | 15.18ms |
| 2 | 0.18/0.31 | 0.74/1.52 | 0.40/0.83 | 8.23ms |
| 3 | **0.17/0.30** | 0.73/1.50 | **0.38/0.80** | 5.93ms |
| 4 | 0.18/0.30 | **0.71/1.46** | 0.39/0.81 | 4.73ms |
| 6 | 0.18/0.30 | 0.73/1.50 | 0.39/0.81 | 3.56ms |

## 4.2 ROBUSTNESS AGAINST OBSERVATION NOISE

In real-world scenarios, historical trajectories are often imperfect due to tracking errors and incomplete detection. It may contain truncated and broken trajectories, mismatched detection candidates, *etc*. To validate the robustness, we introduce two types of noise to simulate detection and tracking errors, as shown in Table 6, and noisy frames are randomly selected. The test is conducted on the JRDB dataset as it is egocentric and naturally prone to issues when obtaining history sequences from noisy tracklets. To deal with noise type Dropped History, we left pad by repeating the first available frames to buffer the dropped observations, which is akin to a stationary person. Since there is no design bottleneck in our framework that precludes stationary states, we consistently produce high-quality predictions across different noise levels. Empirically, we note that a minimum of 2 moving history frames are needed for our method to capture the motion effectively.

## 4.3 ABLATIONS

We further explore the effects of sampler types and different strides with results reported in Table 7. When replacing the sampling from a standard normal distribution with purposive sampling, the performances improve from around 7% to 20% on different datasets. Even with Stage 1 solely, we can achieve comparable prediction performances with other baselines. Experiment results also validate that the Markovian assumption of human motion usually holds up to defined observation frequency intervals with a stride $\tau$ to reach the best performance.

## 4.4 GROUP REASONING VIA POTENTIALS

Potentials learned in the trajectory prediction network can also be utilized in group inference. Leveraging group labels from the JRDB-Act (Ehsanpour et al., 2022) dataset, we perform group clustering without prior assumptions or maximum group number constraints. Concatenated with history image features extracted via CLIP (Radford et al., 2021), we build a binary edge classifier supervised by ground truth groups. The resulting group inference, portrayed in Figure 6, counts individuals as part of a group if there is an edge between them and at least one other agent within the current group. Despite some errors, our method effectively captures scenario relationships, beneficial for pedestrian-friendly navigation tasks adhering to social norms.

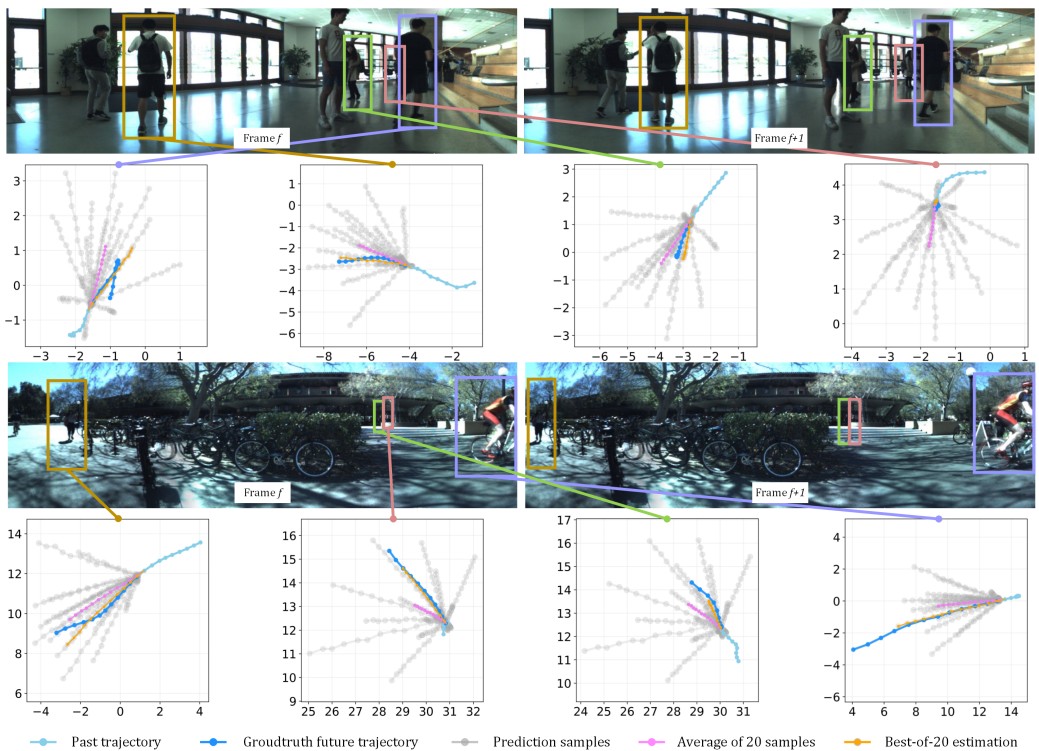

Figure 5: Generated prediction samples with $N = 20$ on JRDB dataset. Best-of-20 samples are highlighted with orange curves.

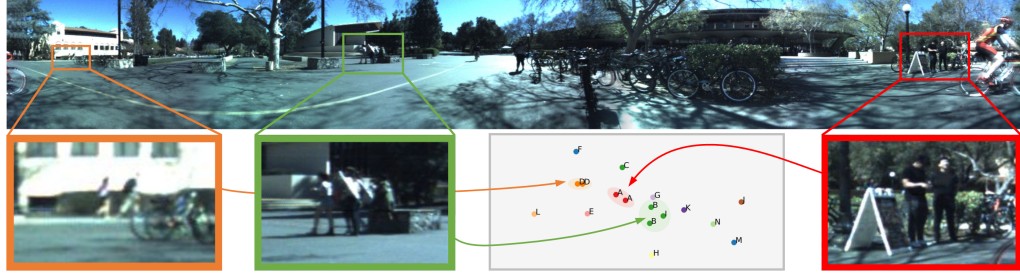

Figure 6: Group inference with learned potentials for JRDB dataset. Matching capital letters represent identical group labels, while ellipses of the same color denote estimated groups. Errors may arise due to low-quality visual cues (*cf.* Person I is misassigned to group B in the green ellipse).

## 5  CONCLUSION

In this paper, we propose a neuralized Markov random field (MRF)-based method for human trajectory prediction. Specifically, we introduce an interaction-aware MRF that models agent motion and crowd interactions over time. We then design a neural network framework consisting of two CVAEs to approximate the posterior distribution for efficient learning and inference. Our method achieves state-of-the-art performance across four benchmark datasets and also enables group reasoning. Furthermore, it is robust against noisy observations and allows for real-time stochastic inference, demonstrating its feasibility on real-world downstream systems.

**Limitations.** We learn the potential function applied to the MRF graph from data; if the training data only exhibits low graph complexity, the model's ability to capture complex interactions will be limited. This complexity generalization issue can be further explored in the future. Moreover, the current prediction pipeline uses only coordinates as input; incorporating environmental contexts, such as obstacles and traversable paths, could also be considered.

ACKNOWLEDGMENTS

This research / project is supported by Agency for Science, Technology & Research (A*STAR), Singapore under its National Robotics Program (No. M23NBK0053), and the National Research Foundation (NRF) Singapore, under its NRF-Investigatorship Programme (Award ID. NRF-NRFI09-0008).

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

## A  APPENDIX

We attach network design details for each sub-component in CVAEs, and data-specific hyperparameters for training and inference in Table 8 and Table 9, respectively. In the training phase, we set hyperparameters $\alpha$, $\beta$ and $\lambda$ in loss terms as $1$, $1 \times 10^{-3}$ and $1 \times 10^{-4}$.

Table 8: Network design details.

| Module Name | Components |
|---|---|
| History Encoder | Conv-1d ($6 \rightarrow 32$) + GRU ($32 \rightarrow 256$) |
| Update Decoder | MLP ($256{+}16 \rightarrow 512 \rightarrow 512 \rightarrow 256 \rightarrow 2{\times}$stride) |
| Configuration Encoder | MLP ($2{\times}$stride $\rightarrow 32 \rightarrow 32 \rightarrow 32$) |
| Dynamics Decoder | MLP ($32{+}16 \rightarrow 128 \rightarrow 128 \rightarrow 64 \rightarrow 2{\times}$stride) |
| Potentials Update | MLP ($3 \rightarrow 16 \rightarrow 32 \rightarrow 32$) |
| History Sampler | MLP ($256 \rightarrow 128 \rightarrow 512 \rightarrow 16{\times}$N) |
| State Sampler | MLP ($32 \rightarrow 32 \rightarrow 64 \rightarrow 16$) |
| Future Encoder | MLP ($2{\times}$stride $\rightarrow 64 \rightarrow 256 \rightarrow 256 \rightarrow 2{\times}16$) |

Table 9: Hyperparameters for different datasets.

| Dataset | Stride | Batch Size | Learning Rate | Step Size | Gamma | Epoch (CVAE) | Epoch (sampler) |
|---|---|---|---|---|---|---|---|
| ETH | 3 | 64 | $2 \times 10^{-4}$ | 16 | 0.5 | 200 | 60 |
| HOTEL | 3 | 50 | $2 \times 10^{-4}$ | 16 | 0.5 | 200 | 60 |
| UNIV | 3 | 32 | $2 \times 10^{-4}$ | 32 | 0.9 | 200 | 60 |
| ZARA1 | 3 | 32 | $2 \times 10^{-4}$ | 32 | 0.9 | 200 | 60 |
| ZARA2 | 3 | 32 | $2 \times 10^{-4}$ | 32 | 0.9 | 200 | 60 |
| SDD | 3 | 16 | $8 \times 10^{-4}$ | 32 | 0.9 | 500 | 200 |
| NBA | 10 | 32 | $1 \times 10^{-4}$ | 32 | 0.6 | 150 | 50 |
| JRDB | 3 | 16 | $1 \times 10^{-4}$ | 32 | 0.6 | 200 | 50 |

In Figure 7, we also compare the initialization and final prediction results for our proposed model and the previous state-of-the-art model LED (Mao et al., 2023) on the NBA dataset. LED starts from a noisy initialization, and it can ignore other candidates when optimizing via the best sample, which results in a distribution with multiple outliers as shown in Figure 7 (c). Conversely, our method produces trajectory-like samples initially due to the modeling of crowd motion and interaction, and converges to a more plausible distribution around the ground truth after optimization.

Furthermore, more visualization results on ETH/UCY datasets for baseline methods NPSN (Bae et al., 2022b), SingularTrajectory (Bae et al., 2024b), and our proposed model, are provided in Figure 8.

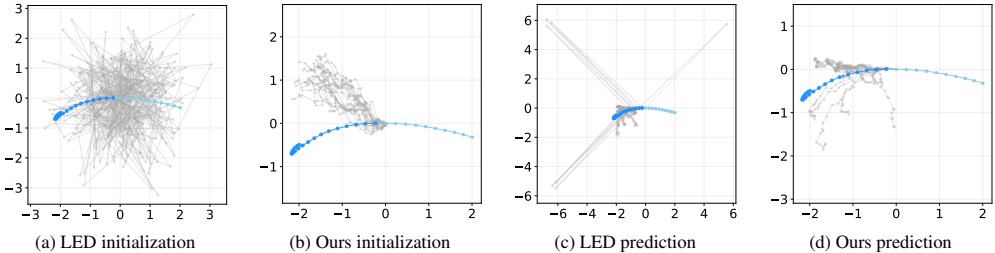

(a) LED initialization    (b) Ours initialization    (c) LED prediction    (d) Ours prediction

Figure 7: Initialization and prediction results of LED and our method. Trajectories are normalized so that the coordinates of the last frames are located at the origin.

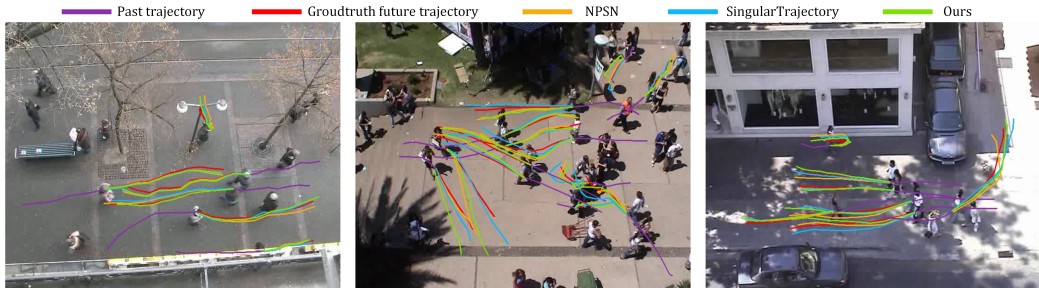

Figure 8: Visualization of the best predictions among generated samples ($N = 20$) with NPSN, SingularTrajectory, and our method.

