# OpenReview forum: "Neuralized Markov Random Field for Interaction-Aware Stochastic Human Trajectory Prediction"
_ICLR.cc/2025/Conference — ICLR 2025 Poster_

### Official Review · Reviewer_m1hq · 2024-10-31

**Soundness:** 3
**Presentation:** 3
**Contribution:** 2
**Rating:** 5
**Confidence:** 3

**Summary:**

Modeling interactions and continuously changing intentions in trajectory prediction is challenging.
To address this, the paper proposes a Markov Random Field (MRF)-based motion evolution method. The trajectory is divided into time chunks, with the assumption that each trajectory segment follows a Markov chain sequentially.
The authors derive crowd motion evolution as a probabilistic distribution, which consists of a Bayesian update term, predicting the next state from each given observation, and a transition term. The transition term is further divided into a self-evolution term, which predicts the next state from each agent’s current state, and an interaction term, modeling relations between agents.
All of these terms are implemented using Conditional Variational Autoencoders (CVAEs).
Experiments are conducted extensively on multiple pedestrian trajectory datasets, achieving state-of-the-art prediction performance and real-time inference speed.

**Strengths:**

* The paper is easy to follow, and the proposed method is technically well-implemented.
* The method is validated through extensive benchmark experiments. Its fast inference speed and high accuracy on the NBA and JRDB benchmarks are impressive, especially in JRDB-based experiments from an ego-centric view where noisy input was tested, and the prediction results appear plausible.
* The experiment applying the proposed method to group reasoning is intriguing, and the visual results are very plausible.

**Weaknesses:**

* It seems difficult to claim state-of-the-art prediction performance. There already are some prediction models showing better performance on ETH-UCY and SDD than the baselines compared here. More convincing reasoning is needed to explain why the proposed method is necessary compared to these models.
    - View Vertically (ECCV’22): 0.18/0.28, SICNet (ICCV’23): 0.19/0.33, Socialcircle (CVPR’24): 0.17/0.27  (: ADE/FDE on ETH-UCY benchmark)
* I find it hard to fully grasp the novelty of the proposed method. As mentioned in the related work, there are already trajectory prediction methods that use the Markov property. Additionally, there are existing methods that construct a dynamic graph based on agents' positions to account for interactions between them (e.g., HiVT (CVPR’22), QCNet (CVPR’23)). The proposed approach seems like a combination of these two approaches, which might not meet the standards for ICLR.
* I initially thought that using CVAE would introduce stochasticity, but it’s odd to see deterministic samplers are used during stage 2. In that case, why split it into two stages? Couldn’t deterministic sampling alone without CVAE sampling be trained and used for inference? The referenced Non-probability Sampling (Bae, CVPR’22) also suggests that deterministic sampling is preferable to random sampling, so I don’t understand why both random and deterministic sampling were used here. The ablation only compares with stage 1 alone; ablation results of state 2 only without CVAE sampling are needed.

**Questions:**

* On line 465, what does "precludes stationary state" mean? How is that related to robustness against noisy input?
    - In addition, perturbation from Gaussian noise doesn’t seem related to stationarity.
* The model uses CVAE in many parts, and CVAE is known to be vulnerable to posterior collapse in autoregressive methods, which could weaken stochastic prediction. What solutions were implemented to address this issue?

---

> ### Author Response · Authors · 2024-11-21
>
> > SOTA performance
>
> There are multiple papers claiming the SOTA performances with their proposed methods when applying the task settings in Social-GAN (CVPR’18), however, the comparison can be **unfair** since they **directly choose the best model based on the test set** instead of following the original train-val-test splits. This issue was mentioned in some GitHub repos (i.e., MID (CVPR’22) issue 5) and papers (i.e., NPSN (CVPR’22)). Therefore, we include the baselines that clearly adhere to the train-val-test splits, for example, AgentFormer (ICCV’21), SingularTrajectory (CVPR’24), or re-produce results with the correct splits setting.
>
> For the baseline mentioned, View Vertically (ECCV’22) or SocialCircle (CVPR’24) also have the same *unfair comparison* issue, which can be checked from the plist files for the dataset in the released codes:
> * V2-net has {'test': ['eth'], 'train': ['hotel', 'zara1', 'zara2', 'univ', 'zara3', 'univ3', 'unive'], 'val': ['eth']};
> * SocialCicle has {'anntype': 'coordinate', 'dataset': 'ETH-UCY', 'dimension': 2, 'scale': 1.0, 'scale_vis': 1.0, 'test': ['eth'], 'train': ['hotel', 'univ', 'zara1', 'zara2', 'univ3', 'unive', 'zara3'], 'type': 'meter', 'val': ['eth']}.
>
> This means when reporting results on the eth subset, they directly choose the best model based on the test set, instead of following the original leave-on-out subset method: train-val on 4 subsets and test on the remaining subset (from Social-LSTM (CVPR’16) and Social-GAN (CVPR’18)).
>
> To have a fair comparison, we also correct the splits used in the most recent SOTA method SocialCicle (CVPR’24), and attach the results here:
> |               |     ETH     |   HOTEL  |    UNIV   |   ZARA1  |  ZARA2 |   AVG   | Speed (57 per.) |
> | :--------------- | :-----------: | :-----------: | :----------: | :----------: | :---------: | :--------: | :-------: |
> | SocialCircle | 0.27/0.42 | 0.13/0.16 | 0.29/0.51 | 0.19/0.33 | 0.14/0.25 | 0.20/0.33 | ~17ms |
>
> It also supports our proposed method is SOTA **under correct train-val-test splits**. For the mentioned SICNet (ICCV’23), it is hard to validate whether the correct splits have been used since no official codes have been released, but our model still surpasses its reported performance on some subsets in ETH/UCY and on SDD (8.44/13.65 compared to ours 7.20/11.29).
>
> In addition, we also show our model outperforms other baselines on the NBA and JRDB datasets, where data splits also strictly follow the prior works. Therefore, we finally claim the state-of-the-art prediction performances across multiple datasets.
>
> > Novelty of the method
>
> Our novelty lies in an explicit modeling on full dynamics of joint space state transitions and crowd interactions with the MRF by a probabilistic distribution, and a neuralized realization for tractable learning and inference. We decompose the full dynamics into the observation model and the dynamics model, imposing a structure prior, unlike other methods that implicitly capture pedestrian motion and interaction mostly in an end-to-end paradigm.
>
> The Markov property implemented in prior works is for *agent-wise self-motion modeling* to show that a pedestrian’s future state (the state can be expressed in either coordinates or abstract action) primarily depends only on its current state, so our proposed method differs starting from the problem formulation, considering the Markovian assumption for segment-wise evolution and in-segment interaction with MRF modeling.
>
> Also, dynamic graph construction itself is a widely applied idea to model interactions as we mentioned in Related Works (c.f. Section 2 Interaction Modeling). However, instead of applying graph networks or cross attentions directly based on distance features and learning the pedestrian’s interaction implicitly in an end-to-end paradigm, we propose explicit modeling of *full dynamics* involves spatial and temporal evolution of the system for all agents and their interactions via decomposed observation model (c.f. Bayesian update) and dynamics model (c.f. self-evolution + interaction), thus benefits from the structured system modeling and neuralized implementation, finally achieving better performances.

---

> > ### Author Response · Authors · 2024-11-21
> >
> > > Stochastic sampler and deterministic sampler
> >
> > Stage 1 is a prototype validation as it directly corresponds to the formulation in Equation 1, we keep it mainly for two reasons: introduce stochasticity that can help with the training of encoders and decodes, and provide number-adaptable sample generation since purposive sampling can only generate trajectories with a fixed number which is specified in the training. Thank you for pointing out the weakness in the ablation, we include the ablation results for applying deterministic sampling only here and also in the updated manuscript, $minADE/FDE_{20}$ are reported:
> > |               |     ETH     |   HOTEL  |    UNIV   |   ZARA1  |  ZARA2 |   SDD   | NBA |
> > | :---------------- | :-----------: | :-----------: | :----------: | :----------: | :---------: | :--------: | :-------: |
> > | Stage 1 Only | 0.28/0.46 | 0.13/0.21 | 0.32/0.58 | 0.20/0.36 | 0.16/0.30 | 8.16/13.37 | 0.83/1.16 |
> > | Stage 2 Only | 0.27/0.40 | 0.11/0.18 | 0.28/0.48 | 0.20/0.33 | 0.15/0.26 | 7.83/12.59 | 0.77/0.98 |
> > |  Two Stages  | 0.26/0.37 | 0.11/0.17 | 0.28/0.49 | 0.17/0.30 | 0.14/0.25 | 7.20/11.29 | 0.75/0.97 |
> >
> > With a small sample number (usually $N$=20 in human trajectory prediction task), applying a deterministic sampler would be better compared to a random sampler, but the stochasticity in the training can also contribute to better performance for the final model, which is a combination of two stages.
> >
> > In addition, as mentioned in Non-probability Sampling (Bae, CVPR’22), generated trajectories are biased toward random sampling, and the bias vanishes when the sampler number goes to infinity. We separate the final model into two stages and keep both, where the deterministic sampler deals with better performance when sample number $N$ is small, and the random sampler is for adaptable or large sample number cases (as we can generate 500 samples for each person under a scenario with 57 agents within 35ms, or 1000 samples within 59ms) without re-training the entire model. For example, one can only train a different sampler under different sample number settings or can utilize the random sampler to generate a large number of samples to mimic a distribution that can be used by downstream robot navigation tasks if applicable.
> >
> > > Additional questions
> >
> > In Table 6 we include two noise types, one is Gaussian Noise for location perturbations, and another is Dropped History which history observations are less than desired frames (9 frames for the JRDB dataset). As explained in L462-464 “we left pad by repeating the first available frames to buffer the dropped observations, which is akin to a stationary person”, stationary states are referred to such instances with noise type Dropped History. Our model is robust under Dropped History noise, as we stated in L464-465.
> >
> > In the training process of CVAE, we apply the annealing schedule trick proposed in *Generating Sentences from a Continuous Space* (Bowman, ICLR’16 Workshop), which is verified effective in preventing posterior collapse. The scheduler is used to linearly increase $\beta$ from 0 to the final hyperparameter setting 0.001 during training.

---

> > > ### Comment · Reviewer_m1hq · 2024-11-26
> > >
> > > Thank you for your responses to my questions.
> > >
> > > Firstly, I find your explanations regarding the SOTA performance and the CVAE posterior collapse convincing.
> > >
> > > Regarding the novelty of your approach, the idea of "crowd interactions with the MRF by a probabilistic distribution, and a neuralized realization for tractable learning and inference" seems closely related to concepts used in "Leveraging Future Relationship Reasoning for Vehicle Trajectory Prediction (ICLR'23)".
> > > I believe it would be good to include a proper citation to this work.
> > > If this is addressed, I think my concerns about the novelty would be somewhat resolved.
> > >
> > > Regarding the additional experiments on the deterministic sampler, the performance difference between the "State 2 only" approach and the proposed two-stage method appears to be somewhat marginal.
> > > However, combined with the reasons provided for its benefits, such as variable number sampling and large number sampling, I find the approach reasonable.

---

> > > > ### Author Response · Authors · 2024-11-28
> > > >
> > > > Thank you for providing this interesting work for discussion.
> > > >
> > > > This *Future Relationship* (ICLR’23) work does not model state dynamics. It computes inter-agent proximity from waypoint occupancy estimated from past trajectories, using it as features to derive interaction edges and features for future prediction. However, there is no state dynamics modeling based on interactions. In contrast, our work introduces explicit modeling of full dynamics through a decomposed observation model (c.f. Bayesian update) and dynamics model (c.f. self-evolution + interaction), iteratively predicting the next state given the current state with the interaction dynamics. Empirically, our dynamics modeling with structure prior achieves better performance compared to previous baselines.
> > > >
> > > > To illustrate this, consider a detailed example: predictions are typically made for a future horizon of 4-6 seconds. When the ego-agent (the prediction target) executes its motion and moves, say, for 2 seconds, other agents also move and respond accordingly. This results in a different spatial relationship compared to the moment when the prediction was initially made (i.e., at 0s). Consequently, the interactions evolve over time, influenced by the agents’ possible motions. This dynamic nature of interactions may not be as apparent in vehicle prediction since vehicles follow lanes and rarely perform sharp maneuvers over short periods. For instance, if two vehicles try to enter the same lane at an intersection, their predicted lane occupancy and interaction features might not vary significantly between time steps 2s and 4s. However, humans exhibit much greater flexibility, making their movements and interactions more dynamic and variable over time.
> > > >
> > > > Therefore, two important characteristics need to be considered for human trajectory prediction: i) Humans can move more freely (in non-driving scenes) and the possible trajectory cannot be enumerated (i.e. relying on lane/path information for coarse future trajectories is insufficient). ii) Humans exhibit greater flexibility in interactions due to their ability to change intentions and adapt quickly, leading to behaviors such as sudden stops, sharp turns, or close detours. These characteristics result in interactions that can change dynamically over time.
> > > >
> > > > It is worth noting that this novel *full dynamics* model is more suited for human agents, as their future trajectories are less constrained and their interactions are more flexible compared to vehicles.
> > > >
> > > > We will cite and thoroughly discuss this work in our final version, and thank you for bringing up this paper.

---

### Official Review · Reviewer_rTU5 · 2024-11-02

**Soundness:** 2
**Presentation:** 3
**Contribution:** 2
**Rating:** 6
**Confidence:** 2

**Summary:**

This paper presents a neutralized Markov random field-based motion evolution method for stochastic human trajectory prediction, which explicitly models the agent’s motion dynamics and crowd interactions, with lightweight and efficient learning as well as inference. Experimental results show that the proposed method is effective on multiple datasets, with robustness under noise disturbance.

**Strengths:**

1. The paper proposes to use MRF and iteratively infer the stochastic distribution, which is different from existing explicit structured models.

2. The numerical results are good, and the inference speed is promising.

3. The paper is clearly written and easy to follow.

**Weaknesses:**

I put both of the Weaknesses and Questions here:

1. The authors should elaborate more on why MRF that iteratively infers the stochastic distribution of future motions could achieve better performance, especially compared to existing methods.

2. It appears to me that this paper clearly express what it does and how it does, but the motivation behind is not very clear, especially when readers are not in the same field, which makes it somewhat confusing why each proposed component would work.

3. I think more visualized results of more baselines and the proposed method are needed for better comparison.

**Questions:**

See Weakness.

---

> ### Author Response · Authors · 2024-11-21
>
> > Motivation and why MRF could achieve better performances
>
> To deal with noisy observations or other uncertainties, the trajectory prediction problem can be formulated as solving a probabilistic distribution of the joint space configuration, in addition to a deterministic function map (c.f. L162-163). Since human self-intention will change and they are constantly interacting with each other, the pedestrian-interaction system dynamics is hard to obtain, and many prior works apply an end-to-end paradigm to learn the dynamics implicitly via different network designs. The main **motivation** for our work is the combination of structured modeling and neural networks, which is inspired by the performance improvements introduced by combining structured modeling and neural networks from other applications, i.e., semantic segmentation (ICCV’15) and stereo matching with MRF (CVPR’24). Therefore, we propose a novel formulation for the trajectory prediction task, explicitly modeling the full dynamics of joint space state transitions and crowd interactions.
>
> We begin with the formulation of the human trajectory prediction task (c.f. L51-72 and Section 3.1) based on the Markovian motion assumption. Then, to capture the dependencies of pedestrians that are constantly interacting, we further factorize the spatial space distribution and propose an MRF-based evolution term (c.f. L72-81), finally realizing tractable learning and inference (Section 3.3) with a new proposed network architecture.
>
> MRF is a very common tool to represent dependencies in the format of a factored probability model specified by undirected graphs, thus is well-suited for relationship modeling for interacting targets. Our MRF modeling takes advantage of the inductive bias in the downstream task, i.e., Markovian assumptions between segments and spatial relations in the graph construction. This structured modeling introduces important spatial prior for interactive agents, and reduces the difficulty for optimization, thus generating plausible samples even in the initialization (c.f. Figure 7 in the appendix where we compare our initializations with LED (CVPR’23), the SOTA model on the NBA dataset). Besides, with neuralized implementation, our MRF modeling overcomes the computational intractability, and learns the distribution from the data, instead of relying on hand-crafted probability functions.
>
> > More visualized results
>
> Thank you for the suggestions and we have updated more visualizations in the appendix of the manuscript on ETH/UCY datasets.

---

> > ### Comment · Reviewer_rTU5 · 2024-11-26
> > **reviewer reply**
> >
> > I appreciate the clear discussion provided by the reviewer, which have addressed my confusion. To be honest I am not familiar with this field, but after reading other reviewers' opinions and author replies, it appears to me that the authors have addressed the raised problems. Anyway, considering the pros and cons of reviewer o4s5, I decide to change my rating to 6.

---

### Official Review · Reviewer_cAi6 · 2024-11-03

**Soundness:** 2
**Presentation:** 2
**Contribution:** 3
**Rating:** 5
**Confidence:** 2

**Summary:**

This work introduces a multi-person tracking approach. It summarizes per-frame tracks into short segments and uses a MRF to forecast those segments, rather than per-frame tracks. To initialize the first trackless, a VAE based “Bayesian update” is performed. Surprisingly, only this first step requires knowledge of the input motion, while the MRF factors out any of the input conditions. The method produces SOTA results on some benchmarks while being significantly faster than most other SOTA methods.

**Strengths:**

The method produces SOTA results on standard benchmarks while greatly outperforming SOTA on runtime too.

**Weaknesses:**

I have two main concerns: [A] the method description is unclear, and [B] the two-stage approach is not well-motivated and not well-ablated.

[A] I find the data description in LL158-161 confusing: could the authors clearly state what the input modalities are, i.e. are those per-person trajectories in 2D, are they collections of 2D per-person trajectories, how are they normalized/standardized - in what coordinate frame are they?
This also makes it difficult to follow the rest of the method description as it is unclear what the inputs/outputs are.
The authors make repeated use of the word “joint configuration space” (i.e. L165) - could they elaborate what that means?

L161: what are those “M individual states”?

The authors use various words to describe there tracklets and the length, i.e. “stride”, “chunk”, “period”, “time chunk” - I would suggest to stick to one to make it easier to read.

L182-L183: It is unclear here what the parameterizations are, i.e. what are the differences between \theta_u, \theta_m and \theta_j —> this needs more details here and not just in the following chapters (Maybe also reference Figure 2 here..).

Can the authors clarify what they mean by L070: “Human movements are Markovian up to certain frequencies” ?



[B] Concerns wrt Two-stage approach
I wonder if the split into Stage 1 (Bayesian Update) and Stage 2 is necessary. My main concern is that Stage 2 has no information about motion of the past beyond the previous segment S_{t-1}), while the segments seem to be rather short, as indicated in Table 7 (right side). I find it surprising that the method does not need to rely on past motion beyond a few frames to make more accurate predictions.

For example, just recursively applying the “Bayesian update” step (Stage 1) should perform better, as it has access to more historic information. I believe this is what the authors evaluated in Table 4 and 5 (“Ours (Stage 1)”) - could the authors confirm that this is indeed the case? They should more clearly describe this baseline / ablation in text. Also, it is surprising that this performs worse than the MRF as this method can exploit past motion better than the MRF. Could the authors elaborate what causes the Bayesian update step to be outperformed here?

On a similar note, if Stage 2 (self-evolution + interaction) seems to perform better, I wonder if the Stage 1 is even necessary as the initial Segment S_1 can be obtained from the historic motion sequence. I wonder how “Ours (Stage 2 only)” would perform in this case.

A two-stage approach is more complicated and brittle and thus it is important to clearly motivate each part, which has not been done in the ablation.

*Suggestions*

Add relative speeds to Table 3 where the fastest method is 1 and the others are multiples of it.

For the stride size in Table 7 (right) I would suggest to add the time in seconds as well

**Questions:**

In Figure 5 the method seems to predict left/right off-shoots - can the authors comment on why those are happening? They seem like very unlikely predictions.

L472: what is a “standard normal distribution”?

---

> ### Author Response · Authors · 2024-11-21
>
> > Input modality, joint configuration space, and parameterization
>
> As introduced in L158-161, $O_t$ is the observations at time $t$ which is a collection of all agents. In the default setting of the human trajectory prediction task (c.f. classical Social-LSTM (CVPR'16), Social-GAN (CVPR'18) methods), $O_t$ is a set of xy-coordinates, which means each datapoint represents a person’s world coordinate. Since many recent approaches (i.e., Social-Transmotion (ICLR'24), etc.) also extend the observations with 2D detection bounding boxes, 2D encoded image features, 2D human skeleton key points, etc., we use a general notation $O_t$ instead of restricting it with one specific modality. **In our method, we adhere to the default xy-coordinates input modality**, and it is also shown in Table 4. The pre-processing of observation sequence $O_{1:t}$ also follows the procedures applied by many previous works (i.e., Trajectron++ (ECCV'20), LED (CVPR'23)): within a single frame, coordinates are standardized with the mean position of all agents, and within an agent, coordinates sequences are aligned with the current time $t$ to ensure the relative position at $t$ is (0, 0). Usually, absolute positions and relative positions are concatenated together and input into the neural network.
>
> Configuration space is usually used to describe the state of a whole system as a single point in a high-dimensional space or to describe assignments of a collection of points to positions in a topological space. We apply this word to describe the pedestrian-interaction system state at a time step (c.f. $X_t$), which includes all agents' spatial coordinates and geometric relationships. In each position we mentioned joint configuration space, there is an equation followed: L160 $X_{t+1:t+T} \triangleq \{X_{t+1}, X_{t+2}, ..., X_{t+T}\}$, L161 $X_t  \triangleq \{X_{i,t}\}^M_{i=1}$, and L165 $S=\{S_1, …, S_K | S_k  \triangleq X_{t+(k-1)\tau+1:t+k\tau}\}$. And $M$ individual states simply mean the states (in trajectory prediction task, spatial arrangements) of all $M$ agents within a time step.
>
> Thank you for the suggestions and we added a reference to the figure for a better explanation of parameterization. The parameterizations in L181-183 are enumerated for different probabilistic function components in Equation 1, thus we include the details of them in Section 3.3 Network Architecture which is the neuralized implementation of each decomposed function, instead of expanding them right after Equation 1.
>
> > Two-stage approach
>
> > L472 standard normal distribution
>
> First, I think there could be some misunderstandings about the two stages here. In Table 4 and 5, “Stage 1 Only” refers to *CVAE Training* as shown in Figure 3. And Stage 2 is *Sampler Training*. “Bayesian Update” and “MRF-based Evolution” are two sub-modules corresponding to two probabilistic function components “Bayesian update” and “self-evolution + interaction” in Equation 1, which is the problem formulation of our method, so they will be both included in Stage 1 and Stage 2.
>
> The difference between the two stages is the sampler. That’s why we mentioned in L472 “replacing the sampling from a standard normal distribution with purposive sampling,” since the standard CVAE structure generates samples from the standard normal distribution $\mathcal{N}(0, I)$ (sampler in Stage 1), and in Stage 2 we further trained a sampler to realize purposive sampling with other encoders and decoders frozen. Stage 1 can be treated as a prototype validation as it is directly corresponding to the formulation in Equation 1. This procedure enables more stochasticity in trajectories. In Stage 2 we apply purposive sampling after the encoders and decoders are trained to make samples generated are dependent on the past, thus allowing more information embedded and that’s the reason for the case after applying Stage 2 the performance will be better.
>
> Besides, in Equation 1 the probabilistic distribution is factorized into “Bayesian update” and “self-evolution + interaction” terms, which can be seen as the observation model and system dynamics model, and the iterative product is from Markovian assumption. While in many prior works, these two models are usually not distinguished and learned implicitly in an end-to-end manner, we are inspired by the improvements introduced by combining structured modeling and neural networks from other applications, i.e., semantic segmentation (ICCV’15) and stereo matching (CVPR’24), then propose a novel formulation for the trajectory prediction task.

---

> > ### Author Response · Authors · 2024-11-21
> >
> > > Human movements are Markovian up to certain frequencies
> >
> > The observation sequence of human motions has its initial sample frequency originating from sensors (i.e., camera, Lidar) and post-processing methods (i.e., up/down-sampling), however, it might not satisfy the Markovian assumption if this frequency is too high. For example, in sports games (like data points in the NBA dataset), the player’s action for the next time step can also depend on the previous states since human response time and muscle reaction are much slower than the frequency of the camera. Therefore, if the future evolution of human movements is independent of its history, we need a frequency that cannot be very high. That’s also the reason that we separate the original sequence into several segments with period $\tau$ (and $\tau$ is a hyper-parameter in the network implementation) so that the system’s transitions are Markovian between those segments.
> >
> > > Explanations for Figure 5
> >
> > Since human intention will change and they are constantly interacting with each other, the future trajectory of a single person is *multi-modal*, highly dependent on self-intention and mutual influence among neighbors (c.f. Figure 4 for the NBA dataset). The prediction samples in Figure 5 show the multi-modality representation ability of our proposed model. For example, the agent highlighted with the pink rectangle in the 1st row might have multiple future actions: stop to join the discussion, directly go through crowds and walk away, go back to avoid crowds, etc. The 4th column in the 2nd row corresponds to this agent, where the color around the center is deeper, revealing the model tends to predict the agent will be stationary. However, we keep the possibility that the agent could move away. Compared to that, the 4th agent in the 4th row has a more compact distribution, since there are fewer bystanders to interact and the intention is clearer.
> >
> > Besides, if considering the standard deviation of the distance errors, we have 0.40/0.78 compared to 0.56/0.96 from the second-best model LED (CVPR’23)
> >
> > > Suggestions on Table 3 and Table 7
> >
> > In Table 3 we tested and included the speeds after GroupNet (CVPR’22) since all the previous baselines in the table didn’t officially demonstrate their results on the NBA dataset, thus there are no official configurations or codes for this dataset. We refer to LED (CVPR’23) for trajectory prediction results and finally put NA (Not Applicable) for other baselines on speed for fairness. Besides, from GroupNet (CVPR’22) the prediction errors are much smaller than previous methods, so we emphasize more on recent methods.
> >
> > Thank you for the suggestion and we attach the inference speed for a scene with 11 persons in the ZARA1 subset here, these results are also updated in the paper:
> > |          Stride        |        1       |        2      |       3       |        4      |       6       |
> > | :-------------------- | :-----------: | :----------: | :----------: | :----------: | :----------: |
> > | Speed (11 per.) | ~15.18ms | ~8.23ms | ~5.93ms | ~4.73ms | ~3.56ms |

---

> > > ### Comment · Reviewer_cAi6 · 2024-11-21
> > > **clarifications**
> > >
> > > Thanks a lot for clarifying my questions:
> > >
> > > **frequency**: I still do not understand what you mean here by "frequency" and what this would have to do with the input modality or with the post-processing steps - Furthermore, I think posing this as Markovian at all is ill-posed: especially for sports, a lot of motion depends on planning and collaboration, which is inherently non-Markovian. For example, imagine for a specific maneuver a player requires a "setup motion" (.i.e. running in a U-shape or anything expressive and clearly visible from motion data) - after a couple of frames, due to your Markovian "stride", the player would forget the maneuver as they have no way of accessing information about it. How this could be handled in this work is still not clear to me.
> > > I am still not convinced by this answer and I strongly encourage to rephrase the claim that "Human movements are Markovian up to certain frequencies".
> > >
> > > **Figure 5**: Thanks for commenting on my question wrt weird off-shoots: I think comparing with SOTA and showing that your method has a lower error variance would support your claim that your method produces less "noisy" samples than other SOTA methods, especially for short time horizons

---

> > ### Comment · Reviewer_cAi6 · 2024-11-21
> > **clarifications**
> >
> > **Issues with model descriptions**: thanks for clarifying the input modalities: - what units are the xy coordinates in - meters, pixels or something else? On the one hand you say that a datapoint represents a "persons world coordinate" (in meters) but then you also say that some methods utilize bbs and 2D skeletons - which would suggest image space. Could you clarify this further? I would suggest to add this to the final version of the paper to make it more self-contained.
> >
> > **configuration space**: I would suggest to call this "joint trajectory space" or "global trajectory space" to be more clear: The use of the word "joint" and missing the input modality description made this stick out, as this wording could be used for skeleton-based approaches as well. I suggest to be more explicit to avoid misunderstandings.
> >
> > **two-stage approach**: thanks for the clarification. My main concern is that the self-evolution process has no access to the "motion history", due to the Markov assumption, so I was wondering if recursively using the Bayesian update would be sufficient, or potentially even stronger: assuming a function exists that $\mathrm{transform}(S_k) \rightarrow O_{t+(k-1)\tau+1:t+k\tau}$ (I assume those are just the 2D trajectories anyways) one could just recursively apply this as a forecasting baseline. The benefit would be that the model always has access to the full history.

---

> > > ### Author Response · Authors · 2024-11-26
> > >
> > > > Issues with model descriptions
> > >
> > > The units are in meters in the world frame, as applied across many datasets. We use the notation $O_t$ for observations as a general representation, allowing the input modality for observations to remain flexible. **Under the default setting, the observation is persons’ world coordinates, but other modalities can be incorporated into the observation if desired** (i.e., combining image space with coordinates, as in some recent methods). In general, including additional modalities in the observation is simply a design choice.
> > >
> > > Thus, in our modeling of $p(X_{t+1:t+T}|O_{1:t}, \theta)$, we use terms like 'states' and 'observations' as general terminology without specifying a fixed modality, as *it can vary based on different implementation design*. In our final design, **we choose to use only xy-coordinates as both the input and output modalities**. We will make it clearer in the revised version.
> > >
> > > > two-stage approach
> > >
> > > Thank you for all the advice on configuration space and approaches. On the two-stage approach, we tried the method that includes the full history as suggested, (i.e., fusing every predicted $S_k$ also into the observation to predict the next $S_{k+1}$, and there are **no** obvious performance improvements compared to our current method. Results are presented in our responses for "Experiments for accessing history" item as well.

---

> > > > ### Comment · Reviewer_cAi6 · 2024-11-26
> > > > **reply to additional comments**
> > > >
> > > > I highly appreciate the additional experiments and that the authors verified that making the model non-Markovian (by giving access to the entire sequence history) does not significantly improve the results on one experiment (ADE/FDE) - however, this experiment only looks at the best result of 20 samples, so there is no indication of how good the other samples are, making it really difficult to actually assess the improvement. How often did you repeat these experiments and could not just the means but also the standard deviations be reported?
> > > >
> > > > Also, I am still not convinced by the explanation of why the Markovian assumption should hold / for short time segments (a few milliseconds) physics / inertia should dominate, which could be approximated with a model under Markovian assumption. However, for longer time horizons of a few seconds, those become less important and long-range dependencies (such as planning) play a more vital role. In this work this logic is reversed: the short-term motion is non-Markovian while the long-term motion is modeled as Markovian.
> > > >
> > > > = = below will not impact my review score, no need to run additional experiments based on this suggestion = =
> > > >
> > > > I think for future works utilizing a multi-modal version of those metrics might be helpful [A,B] as a lot of the trajectories seem very noisy and ADE/FDE in a way cherry-picks from the generations.
> > > >
> > > > [A] DLow: Diversifying Latent FLows, ECCV20
> > > >
> > > > [B] Generating smooth pose sequences for diverse human motion prediction, ICCV21

---

> > > > > ### Author Response · Authors · 2024-11-28
> > > > >
> > > > > Thank you for the further discussion. We choose to report minADE/FED$_{20}$ as they are the most commonly used metrics for evaluating human trajectory prediction tasks across previous baselines. And we agree that these metrics might not fully reflect the overall performance across all samples (as they focus only on the best-case performance, for example in Figure 4, some trajectories generated by LED (CVPR'23) could be very noisy). Therefore, we also present the average errors and the standard deviation of errors on the NBA dataset, similar to what is shown in Table 7:
> > > > >
> > > > > | Time | Original (AVG.) | New suggestion (AVG.) | Original (SD.) | New suggestion (SD.) |
> > > > > | :------ | :------------------: | :-----------------------------: | :------------------: | :---------------------------: |
> > > > > |  1.0s | 0.462/0.920 |    0.465/0.925    | 0.186/0.434 |    0.193/0.428    |
> > > > > |  2.0s | 1.134/2.399 |    1.140/2.431    | 0.524/1.196 |    0.537/1.226    |
> > > > > |  3.0s | 1.862/3.691 |    1.858/3.732    | 0.858/1.869 |    0.883/1.902    |
> > > > > |  4.0s | 2.450/4.548 |    2.456/4.545    | 1.145/2.279 |    1.154/2.276    |
> > > > >
> > > > > We repeated the training and evaluation five times and the metrics remained similar, differing mainly in the third decimal place. Therefore, we conclude that the performance differences between the two are negligible.
> > > > >
> > > > > > Markov assumption
> > > > >
> > > > > As multiple time ranges are involved, we would like to first confirm if the reviewer considers the short-term motion to be Markovian and the long-term not. If that is the case, we agree, as in our last response, “We assume the short-term motions of each agent are primarily influenced by the most recent actions of nearby agents, making Markovian modeling suitable between segments” and “We agree that the long-term reasoning of humans is not Markovian because of external instructions, self-intentions, etc.”
> > > > >
> > > > > Thanks for the suggestions on multi-modal metrics. As the multi-modal version may involve the grouping of similar past motions to obtain multi-modal ground truth, finding proper criteria for grouping history trajectories across various scenarios under constantly changing nearby agents’ interactions could be difficult for the datasets, but it is interesting and valuable to explore.

---

> ### Author Response · Authors · 2024-11-26
>
> > Frequencies and Markovian assumptions
>
> Here using “certain frequencies” we would like to express that the Markovian assumption might not hold under the original video/sensor frequency, so we use the “stride” to divide the sequences into segments, and assume the Markov property between segments.
>
> Each segment $S_k$ contains **all agents and their reciprocal interactions**, thus we include the collaboration between each other. Note that the Markovian assumption we presented in L166-168 is for segments.
>
> > Short-term predictions and Markovian approximations
>
> We agree that the long-term reasoning of humans is not Markovian because of external instructions, self-intentions, etc. (like the example mentioned, the player has the long-term goal to perform a maneuver to reach a target location).
>
> Nevertheless, we assume the **short-term motions** of each agent are primarily influenced by the most recent actions of nearby agents, making Markovian modeling suitable between segments. For example, if a player’s maneuver is blocked by opponents, it is predictable that the player will respond to this (i.e., detour, sudden turns, etc. that can be observed in the data), regardless of their longer-term goal, which admittedly depends on their history.
>
> In the human trajectory prediction task, we usually predict for the future 4-5 seconds based on 3-4 seconds observations in *multi-agents interactive scenarios*, thus more fitting for this assumption. In detail, under our assumption, we first update the future 1-2s trajectories using the observation. Then, trajectories for the further future are predicted based on the Markov property.  We do utilize the history, and the Markovian assumption applies during the further future, representing short-term motions. And we do not handle the hidden intention estimation explicitly in our method.
>
> > Experiments for accessing history
>
> Besides, we also attach the experiment results for the NBA dataset when recursively aggregating the “motion history” for the prediction as suggested (i.e., incorporating prediction as expected history $transform(S_k) \rightarrow O_{t+(k−1)\tau+1:t+k\tau}$):
>
> | Time |     Original    | New suggestion |
> | :------ | :--------------: | :--------------------: |
> |  1.0s | 0.165/0.240 |     0.165/0.241    |
> |  2.0s | 0.337/0.500 |     0.337/0.500    |
> |  3.0s | 0.534/0.745 |     0.533/0.741    |
> |  4.0s | 0.750/0.971 |     0.750/0.975    |
>
> There are *no obvious improvements* when accessing more “motion history”; contrarily, the inference speed will decrease. For other datasets we used, the phenomenon is the same (i.e., on the ETH set, we have the original metrics 0.257/0.374 compared to 0.264/0.381 with full history). It supports our assumption experimentally, and the Markov property can enable more time efficiency.
>
> Thank you for pointing out that the original statement is not very rigorous, and we will rephrase this assumption as **“Human movements are Markovian up to a certain time under local interaction in a short time horizon”** in our manuscript.

---

### Official Review · Reviewer_o4s5 · 2024-11-04

**Soundness:** 3
**Presentation:** 2
**Contribution:** 2
**Rating:** 8
**Confidence:** 5

**Summary:**

The paper proposes a method for human trajectory prediction that incorporates interaction-awareness through a neuralized Markov Random Field (MRF) framework. By integrating MRF with neural networks, the model captures both individual movement patterns and inter-agent interactions over time, effectively addressing uncertainties in predicting human motion. This design is particularly well-suited for real-time trajectory forecasting.

**Strengths:**

- Novel integration of Markov Random Fields with neural networks for human trajectory prediction.
- Advances the field by combining structured probabilistic modeling with deep learning to address noisy, dynamic environments.
- Achieves strong performance on multiple datasets with short inference time.
- Transparent documentation of training and hyperparameter choices ensures reproducibility.

**Weaknesses:**

- Important citations about the dataset used in this paper are missing. Eg., ETH, UCY, Stanford Drone Dataset (SDD)...
- In table 1, it will be interesting to add the previous model Y-net [1] as a baseline since that model had very impressive performance on the datasets used in this paper.
- The robustness experiment is not clear. E.g., when simulating noisy tracklets by adding Gaussian Noise, which 4 frames out of 9 observed frames are selected to add noise? Are they randomly selected or some specified 4 frames? In addition, adding noise to all 9 observed frames is more convincing to simulate sensor noise.



[1] Mangalam, Karttikeya, et al. "From goals, waypoints & paths to long term human trajectory forecasting." Proceedings of the IEEE/CVF International Conference on Computer Vision. 2021.

**Questions:**

- Figure 1 was never referred to in the text. It will be good to mention it in your text.
- In table 4, it seems there is a gap between the proposed method and the previous works on JRDB dataset. Please double-check if there is something wrong. (e.g., same data splits? data samples?)

---

> ### Author Response · Authors · 2024-11-21
>
> > The robustness experiment is not clear.
>
> In Table 6, we randomly select 4 frames from all 9 observed frames to add Gaussian noise. And thank you for the suggestions, we attached the robustness tests where all 9 frames are disturbed by Gaussian noise:
> | Noise Type                                            |  ADE/FDE  |
> | :-------------------------------------------------- | :-------------: |
> | Gaussian Noise (scale=0.1, 9 frames)  |  0.28/0.50  |
> | Gaussian Noise (scale=0.2, 9 frames)  |  0.34/0.57  |
> | Gaussian Noise (scale=0.5, 9 frames)  |  0.88/1.44  |
>
> Usually, a scale of 0.1 is treated as severe noise, and the results in the table show that our model is still robust under noise applied to all frames.
>
> > In table 1, it will be interesting to add the previous model Y-net
>
> We have tried to include Y-net but it is hard to replicate and validate the results since the codes released on GitHub have several problems. First, the configs for Y-net on the ETH/UCY dataset are not provided (c.f. GitHub issue 63) and the codes have some bugs with the dataloader as well (c.f. GitHub issue 67). And there are no pre-trained models for this dataset. Second, based on the data folder released, it also applies an **unfair** comparison where **the best model was also chosen directly based on the test set**, which is different from the standard train-val-test split in Social-GAN (CVPR’18) and is mentioned by the authors of Social-STGCNN(CVPR’20) and NPSN (CVPR’22). We highlight this issue in the caption of Table 1 and finally show the performances of SOTA models that either adhere to the leave-one-out strategy or can be reproduced by re-training with the correct splits.
>
> > It seems there is a gap between the proposed method with the previous works on JRDB dataset
>
> We double-checked and confirmed the task settings and data splits are the same as in previous works. As mentioned in our paper L301-302, we follow the same prediction task settings (i.e. predict for the next 12 time-steps given the past 9 time-steps at 2.5 fps) and data splits with the previous work Social-Transmotion (ICLR’24) to evaluate our model and LED (CVPR’23), the SOTA model on NBA dataset. In detail, *gates-ailab* for validation, indoor scenario series *packard-poster-session* and outdoor scenario series *tressider* for testing, and the other scenarios for training. Besides, the JRDB dataset provides toolkits for tracking tasks on GitHub, and we simply apply official codes to process their original detection labels to generate trajectories within each scenario. For baselines before Social-Transmotion, we follow the results in the Social-Transmotion paper.
>
> In addition, since we have baselines Social-GAN (CVPR’18), Trajectron++ (ECCV’20), and LED (CVPR’23) on both the NBA dataset (c.f. Table 3) and the JRDB dataset, the differences between our proposed model and these baselines follow a similar trend. This phenomenon also supports our implementation.
>
> > Important citations about the dataset used in this paper are missing.
> > Figure 1 was never referred to in the text
>
> Thank you for pointing these out; we have updated them in our manuscript.

---

> > ### Comment · Reviewer_o4s5 · 2024-11-22
> >
> > Thanks a lot for the clarification.
> >
> > Regarding the main experiment on JRDB, the previous baselines treated raw tracking data as trajectories, as the selected scenarios lacked ego-camera movement. Consequently, the comparison in Table 4 may be unfair, given that the data samples used are not consistent across methods.

---

> > > ### Author Response · Authors · 2024-11-26
> > >
> > > As all the other datasets are in the world frame (a consistent reference frame), and the JRDB provides their recorded rosbag files for all scenarios on their website, we transform all tracklets into the world frame for coordinate consistency with the ego-motion extracted from the /tf topic (although it could be noisy as well).
> > >
> > > > Previous baselines treated raw tracking data as trajectories
> > >
> > > We understand the reviewer’s consideration as previous baselines would apply the moving camera frame as the reference (thank you for pointing out since it is not mentioned in the dataset section of Social-Transmotion). We reran our experiments using this setting and here are the results for both the deterministic setting (c.f. Table 4) and stochastic setting (c.f. Table 5):
> > >
> > > (Deterministic) Ours: 0.33/0.63 for Stage 1 Only, and LED (CVPR’23) baseline 0.36/0.69
> > >
> > > (Stochastic N=20) Ours: 0.30/0.46 for Stage 1 Only and 0.27/0.40 for two stages
> > >
> > > It still supports that our proposed model achieves the best performance. And we will add these results to our manuscript.

---

> > > > ### Comment · Reviewer_o4s5 · 2024-12-02
> > > >
> > > > Thanks for the comprehensive clarification and the new updated results.  I will increase my rating to support this work.

---

### Meta-Review · Area_Chair_B2n6 · 2024-12-17

**Metareview:**

The paper proposes a method for human trajectory prediction that incorporates interaction-awareness through a neuralized Markov Random Field (MRF) framework. Reviewers' concerns included lack of citations, unclear experimental details and method descriptions, lack of necessary ablation experiments, and limited novelty.The author solved the above problems during the rebuttal. So the final vote is acceptance.

**Additional Comments On Reviewer Discussion:**

After rebuttal, the reviewers raised their votes to support of this work.

---

### Decision · Program_Chairs · 2025-01-22

Accept (Poster)